# Mechanistic Modelling Identifies and Addresses the Risks of Empiric Concentration-Guided Sorafenib Dosing

**DOI:** 10.3390/ph14050389

**Published:** 2021-04-21

**Authors:** Warit Ruanglertboon, Michael J. Sorich, Ashley M. Hopkins, Andrew Rowland

**Affiliations:** College of Medicine and Public Health, Flinders University, Bedford Park, SA 5042, Australia; michael.sorich@flinders.edu.au (M.J.S.); ashley.hopkins@flinders.edu.au (A.M.H.); andrew.rowland@flinders.edu.au (A.R.)

**Keywords:** concentration-guided dosing, model informed dosing, physiologically based pharmacokinetics, sorafenib

## Abstract

The primary objective of this study is to evaluate the capacity of concentration-guided sorafenib dosing protocols to increase the proportion of patients that achieve a sorafenib maximal concentration (C_max_) within the range 4.78 to 5.78 μg/mL. A full physiologically based pharmacokinetic model was built and validated using Simcyp^®^ (version 19.1). The model was used to simulate sorafenib exposure in 1000 Sim-Cancer subjects over 14 days. The capacity of concentration-guided sorafenib dose adjustment, with/without model-informed dose selection (MIDS), to achieve a sorafenib C_max_ within the range 4.78 to 5.78 μg/mL was evaluated in 500 Sim-Cancer subjects. A multivariable linear regression model incorporating hepatic cytochrome P450 (CYP) 3A4 abundance, albumin concentration, body mass index, body surface area, sex and weight provided robust prediction of steady-state sorafenib C_max_ (*R*^2^ = 0.883; *p* < 0.001). These covariates identified subjects at risk of failing to achieve a sorafenib C_max_ ≥ 4.78 μg/mL with 95.0% specificity and 95.2% sensitivity. Concentration-guided sorafenib dosing with MIDS achieved a sorafenib C_max_ within the range 4.78 to 5.78 μg/mL for 38 of 52 patients who failed to achieve a C_max_ ≥ 4.78 μg/mL with standard dosing. In a simulation setting, concentration-guided dosing with MIDS was the quickest and most effective approach to achieve a sorafenib C_max_ within a designated range.

## 1. Introduction

Sorafenib is an orally administered small molecule kinase inhibitor (KI) used in the treatment of advanced hepatocellular (HCC) and renal cell (RCC) carcinomas. Sorafenib is a potent inhibitor of multiple kinase receptors including the vascular endothelial growth factor receptor (VEGFR), endothelial growth factor (subtype 1, 2 and 3), platelet-derived growth factor-beta (PDGFRβ) and fibroblast growth factor receptor 1 (FGFR1). Variability in sorafenib exposure between individuals and within an individual over time has been identified as a potential source of heterogeneity in treatment efficacy and tolerability [1,2]. The area under the plasma concentration curve (AUC) and maximal concentration (C_max_) for sorafenib following has been reported to vary more than 50% with standard 400 mg dosing [3,4,5]. Variability in gastrointestinal absorption due to limited and pH dependent solubility has been proposed as a major source of variability in exposure [6], however concomitant proton pump inhibitor (PPI) use, which is reported to reduce KI absorption [7], has been demonstrated to have no impact on survival outcomes in HCC [8] and RCC [9,10] patients treated with sorafenib.

A sorafenib C_max_ ≥ 4.78 μg/mL has been associated with superior overall survival in RCC and HCC patients, albeit with a higher incidence of hypertension, while a sorafenib C_max_ ≥ 5.78 μg/mL has been associated with an increased incidence of grade II toxicity, the most common of which is hand foot skin reactions [11,12]. While the evidence for these thresholds is derived from a single observational study in 52 individuals, these values have been cited as target concentrations in multiple reviews addressing individualised sorafenib dosing [13,14]. The dose escalation protocol that has been proposed for sorafenib to increase from 400 mg to 600 mg twice daily [15]. This approach is based on a sub-analysis of a phase II trial demonstrating a clinical benefit in patients who increased from 400 mg to 600 mg sorafenib twice daily following disease progression at the 400 mg dose [16]. Notably the association of this dose escalation with sorafenib plasma concentration has not been evaluated.

The potential benefits of individualised KI dosing have gained interest in recent years [17,18,19] and a number of strategies are available to both inform initial dose selection and facilitate dose adaption [20]. Therapeutic drug monitoring (TDM) is an established method to facilitate concentration-guided dose adaption but requires significant clinical and analytical resources to quantify the drug of interest and establish a robust evidence base. To date, few cancer medicines have met the level of evidence required to implement TDM in a clinical setting [21,22,23]. 

Model-informed initial dose selection (MIDS), often underpinned by a population pharmacokinetic (pop-PK) or physiologically based pharmacokinetic (PBPK) model, has emerged as a strategy to assist initial dose selection either to complement or replace TDM [24,25,26,27]. PBPK modelling and simulation is an established tool in drug discovery and development, where it is used to predict factors affecting PK and support the design of clinical trials [28,29]. PBPK is a ‘bottom-up’ approach whereby the concentration–time profile of a drug is simulated based on physiochemical and in vitro data [30,31]. Novel clinical applications for PBPK have been proposed involving the prediction of clinical drug–drug interactions, identification of physiological covariates impacting drug exposure and informing initial dose selection [25,31,32,33]. 

The primary objective of this study is to evaluate the capacity of concentration-guided sorafenib dose adjustment, with and without MIDS, to increase the proportion of patients that achieve a sorafenib C_max_ within a concentration range of 4.78 to 5.78 μg/mL. A full body PBPK model for sorafenib was first developed and validated, then used to identify physiological and molecular covariates associated with between subject variability in sorafenib exposure.

## 2. Results

### 2.1. Verification of the Sorafenib PBPK Compound Model

The accuracy of the sorafenib compound model was assessed in nineteen age and sex matched cohorts from single or multiple ascending dose (100 to 800 mg) trials. Mean simulated and observed AUC and C_max_ values and the corresponding simulated/observed ratios are presented in Appendix A along with a summary of the verification trial characteristics (i.e., age range, sex, sample size and dose). The mean (±standard deviation; SD) simulated/observed AUC and C_max_ ratios for the single-dose cohorts (*n* = 37) were 1.92 (±1.11) and 1.50 (±1.7), respectively. The mean (±SD) simulated/observed AUC and C_max_ ratios for the multiple-dose (typically 14 days) cohorts (*n* = 14) were 1.50 (±0.72) and 1.17 (±0.63), respectively. Variability in model performance, indicated by large SD for parameter ratios, was driven by heterogeneity in observed parameters between trials. A representative sorafenib concentration–time profile depicting overlayed with the mean concentration–time profile and 90% confidence interval (CI) for the observed data is shown in Figure 1. The accuracy of the sorafenib compound model was considered acceptable on the basis that mean simulated parameters were within two-fold of the respective mean observed parameter and contained within the 90% CI for the observed parameter. Simulated Day 14 and Day 28 sorafenib C_max_ values were divided by 1.17 to account for simulation to observed MFE in multiple dose studies when evaluating the simulated parameters against the observed target C_max_ range. Results of sensitivity analyses performed to evaluate the impact of input parameters with measurement uncertainty (C_lint_ for CYP3A4 and UGT1A9 pathways, fraction unbound, B/P ratio and LogP) on sorafenib kinetic parameters are shown in Appendix A.

### 2.2. Sorafenib Exposure in Cancer Patient

The summary of the mean, SD and range of steady-state sorafenib AUC and C_max_ parameters defining exposure in 1000 virtual cancer patients is presented in Appendix A. Consistent with the reported clinical trial data [4,5,34], the simulation revealed variability of greater than an order of magnitude in sorafenib exposure; the steady state AUC ranged from 22.7 to 270 mg/L·h (mean 99.2 mg/L·h), while C_max_ ranged from 2.3 to 23.2 µg/mL (mean 8.9 µg/mL). 

### 2.3. Physiological and Molecular Characteristics Driving Variability in Sorafenib Exposure

Univariate logistic regression analysis evaluated correlations between physiological and molecular characteristics and sorafenib steady state C_max_ threshold at > 4.78 mg/L (Appendix A) in a cohort of 1000 Sim-Cancer subjects. Statistical analysis of multivariable linear regression with stepwise inclusion of parameters revealed the primary covariates driving variability in sorafenib AUC were hepatic CYP3A4 abundance, albumin concentration, body mass index (BMI), body surface area (BSA), sex and weight (Figure 2). 

A summary of the performance characteristics for the multivariable linear regression model is shown in Table 1. The covariate most strongly associated with variability in sorafenib AUC was hepatic CYP3A4 abundance, inclusion of albumin concentration and BMI resulted in substantial improvement in multivariable model fit. Stepwise inclusion of additional the covariates BSA, sex and weight resulted in minor improvements in model performance (*R*^2^ change ≤ 0.010). No other covariate met the stepwise inclusion criteria (probability of F to enter ≤ 0.05). These parameters formed the basis of the MIDS. The AUC of the ROC for the MIDS predicted steady state AUC was 0.991 (Figure 3). Sixty-three subjects (6.3%) from the Sim-Cancer cohort failed to achieve a Day 14 C_max_ > 4.78 µg/mL. MIDS predicted individuals that failed to achieve a therapeutic sorafenib C_max_ with 95.2% sensitivity (60/63 sub-therapeutic individuals) and 95.0% specificity (809/937 therapeutic individuals) (Table 2, Figure 3). Shown in Appendix A, differences in albumin concentration between participants were associated with changes in f_u_.

### 2.4. Impact of Dose Individualisation

The proportion of participants with a simulated sorafenib C_max_ below, within and above the target 4.78 to 5.78 µg/mL range at Day 14 and Day 28 based on following flat 400 mg dosing, concentration-guided dosing and concentration-guided dosing with MIDS is reported in Table 3. Concentration-guided sorafenib dosing without MIDS identified that 12.4% of subject (62/500) failed to achieve a Day 14 C_max_ ≥ 4.78 µg/mL with 400 mg twice daily dosing. Increasing the sorafenib dose to 600 mg twice daily in individuals who failed to achieve a Day 14 sorafenib C_max_ > 4.78 µg/mL, while retaining the 400 mg twice daily dose for those who did achieve a Day 14 sorafenib C_max_ ≥ 4.78 µg/mL resulted in 99% of subjects (495/500) achieving a Day 28 C_max_ ≥ 4.78 µg/mL. Concentration-guided sorafenib dosing without MIDS resulted in an additional 43 subjects achieving a Day 28 C_max_ > 5.78 µg/mL compared to flat 400 mg dosing.

On the basis of MIDS, 52 subjects were allocated to receive an initial sorafenib dose of 500 mg and 448 subjects were allocated to receive an initial sorafenib dose of 400 mg. Concentration-guided sorafenib dosing with MIDS resulted in 6.8% (34/500) subjects failing to achieve a Day 14 C_max_ ≥ 4.78 µg/mL. Increasing the sorafenib dose to 600 mg twice daily in individuals who failed to achieve a Day 14 sorafenib C_max_ ≥ 4.78 µg/mL, while retaining the MIDS informed twice daily dose for those who did achieve a Day 14 sorafenib C_max_ ≥ 4.78 µg/mL resulted in 99% (495/500) of subjects achieving a Day 28 C_max_ ≥ 4.78 µg/mL. Concentration-guided dosing with MIDS resulted in an additional 9 subjects achieving a Day 28 C_max_ > 5.78 µg/mL compared to flat 400 mg dosing. Post-hoc analysis demonstrated that three of these subjects would have a C_max_ < 4.78 µg/mL with 400 mg dosing, while the remaining six subjects could have retained a Day 28 C_max_ ≥ 4.78 µg/mL while avoiding a Day 28 C_max_ > 5.78 µg/mL with a dose reduction from 500 mg to 400 mg following assessment of C_max_ on Day 14, however dose reduction was not incorporated into the simulation protocol. 

## 3. Discussion

The present study demonstrated that concentration-guided dosing with MIDS facilitates therapeutic sorafenib exposure in 99% of subjects within 28 days while minimising the number of additional subjects at risk of supra-therapeutic dosing compared to concentration-guided dosing alone. Multivariable linear regression modelling demonstrated that variability in simulated sorafenib AUC and C_max_ is associated with hepatic CYP3A4 abundance, albumin concentration, BMI, sex, age and weight. Logistic regression modelling of these covariates predicted individuals likely to fail to achieve sorafenib C_max_ ≥ 4.78 mg/L with high sensitivity and specificity (95.2% and 95%, respectively). Incorporation of these parameters into an MIDS algorithm that allocated subjects to a 400 mg or 500 mg initial sorafenib dose resulted in a 50% reduction in the number of subjects that failed to achieve a Day 14 C_max_ ≥ 4.78 mg/L. When used in conjunction with concentration-guided dosing at Day 14, this protocol resulted in 99% of subjects attaining a Day 28 C_max_ ≥ 4.78 mg/L. 

The current study also highlights the potential danger of empiric concentration-guided dosing in terms of placing patients at an increased risk of toxicity. In the absence of MIDS, 69% of subjects (43/62) that underwent a dose escalation from 400 to 600 mg on Day 14 experienced a C_max_ on Day 28 that is associated with increased risk of grade II toxicity. Compared to concentration-guided dosing alone, the concentration-guided dosing with MIDS protocol reduced the number of additional subjects at increased risk of grade II toxicity on Day 28 (C_max_ ≥ 5.78 mg/L) from 43 to 9.

PBPK modelling and simulation is an established tool to support drug discovery and development, and is a core element of the regulatory approval process in many jurisdictions [35]. Recent studies have further demonstrated the potential role of PBPK in predicting covariates affecting variability in drug exposure resulting from either patient characteristics or the drugs’ physicochemical properties [24,25], giving rise to the intriguing potential for this platform to support model informed precision dosing [26,32]. Since the introduction of imatinib in 2001 there has been a growing evidence base supporting a role for concentration-guided KI dosing, despite this implementation of KI dose individualisation has remained challenging. Many early studies focussed on a potential role for TDM-guided KI dosing, however, sufficient evidence has yet to be generated to support widespread implementation for any KI. This has led to the exploration of novel approaches to facilitate precision KI dosing, which have included model informed precision dosing based on integrated simulation/prediction platforms such as PK-Sim^®^, GastroPlus^TM^, Phoenix^TM^, and Simcyp^®^ [26,36,37,38].

The target concentration range and dose escalation protocol used in the current study were based on the best current evidence [11,16]. The main limitation to this study remains the lack of independent verification of the 4.78 to 5.78 µg/mL target C_max_ range. Further, when considering the clinical implementation, it is also important to note that the rate at which sorafenib is absorbed from the GIT varies >five-fold [6]. Variability in the rate of intestinal absorption results in marked variability in the time taken to reach C_max_ for sorafenib (1 to 6 h). As such, in the absence of full PK (AUC) sampling, which is not practical in a clinical setting, concentration-guided sorafenib dosing based on a C_max_ target is unlikely to be robust.

Liver CYP3A4 abundance was identified as the dominant characteristic driving variability in sorafenib AUC and C_max_. By accounting for this characteristic alone, it was possible to identifying subjects with a sub-therapeutic sorafenib C_max_ with a specificity of 74.6% and a sensitivity of 96.3%. When hepatic CYP3A4 abundance was considered along with readily attained data regarding albumin concentration, BMI, BSA, sex and weight in combination with albumin concentration, these two parameters accounted for >88% of multivariable model performance in terms of *R*^2^, specificity and sensitivity (Table 1). These data suggest that consideration of liver CYP3A4 abundance may provide sufficient power to prospectively identify patients who are likely to require a higher sorafenib dose in order to achieve a therapeutic plasma concentration. Importantly, recent work in this [39] and other [40,41] laboratories has demonstrated that quantification of extracellular vesicle (EV)-derived CYP3A protein, mRNA and ex vivo activity robustly describes variability in CYP3A activity in humans. 

This study identified the major physiological and molecular characteristics associated with between subject variability in sorafenib exposure to be hepatic CYP3A4 abundance, albumin concentration, BMI, BSA, sex and weight. Initial dose selection informed by a model accounting for these covariates resulted a quicker and more effective concentration-guided sorafenib dosing.

## 4. Materials and Methods

### 4.1. Development and Verification of the Sorafenib PBPK Model Structural Model

Sorafenib absorption was simulated using the advanced dissolution, absorption, and metabolism (ADAM) sub-model which incorporates membrane permeability, intestinal metabolism and transporter-mediated uptake and efflux. The ADAM sub-model was used in conjunction with a full-body PBPK model, containing compartments and drug distribution characteristics for all organs. All simulations were performed using Simcyp^®^ (version 19.1, Certara, UK). The differential equations underpinning the model have been described previously [42]. 

### 4.2. Development of the Sorafenib Compound Model

The physicochemical, blood binding, absorption, distribution, elimination parameters utilised to construct the sorafenib compound model are summarised in Table 4. Physicochemical properties were based on published literature and documents [43,44]. Metabolism and elimination parameters were incorporated based on reported intersystem extrapolation factor (ISEF) adjusted in vitro CYP and UDP-glucuronosyltransferase (UGT) data (Figure 4).

### 4.3. Population Model

As no clinical trials evaluating sorafenib exposure have been performed in healthy volunteers, verification of the sorafenib compound model was performed using Sim-Cancer population cohort. Simulations performed to assess the physiological and molecular characteristics driving between-subject variability in sorafenib exposure at steady state also utilised the Sim-Cancer population cohort. The physiological and pathological characteristics of the Sim-cancer population have been determined based on a meta-analysis of cancer patients enrolled in clinical trials [45]. 

### 4.4. Simulated Trial Designs

During the model development stage, simulations included 10 trials with 10 subjects per trial (100 subjects total). During the verification stage, simulations were performed in 10 trials matched for sample size, dose, age range and sex distribution in the protocol described for the observed trial. Unless specified otherwise, parameters defining sorafenib exposure were assessed over 24 h following a single dose at 9:00 a.m. on day 1. 

### 4.5. Validation of the Sorafenib Compound Model 

The sorafenib compound model was validated by comparing simulated AUC and C_max_ values to reported observed values from matched clinical trials undertaken in cancer patients. A mean simulated parameter estimated within two-fold of the mean observed parameter and contained within the 90% confidence interval for the observed parameter was applied as the criteria to accept the model accuracy. The model goodness-of-fit was further verified by visual inspection of the overlay of mean simulated and observed sorafenib concentration-time profiles from individual clinical trials. Simulated C_max_ values were normalised to account for the mean fold error (MFE) between simulated and observed values determined from multiple dose validation studies when evaluating simulations against the observed target concentration range (4.78 to 5.78 µg/mL).

### 4.6. Physiological and Molecular Characteristics Driving Variability in Sorafenib Exposure

The validated sorafenib compound model was used to evaluate associations between physiological and molecular covariates and steady-state sorafenib AUC and C_max_ [24]. A trial comprising 1000 subjects from the Sim-Cancer population was simulated over 15 days with 400 mg of sorafenib administered orally in a fasted-state every 12 h for 14 days starting at 9:00 a.m. on Day 1. The steady state sorafenib AUC was determined over 12 h following the final dose of sorafenib at 9:00 p.m. on Day 14. The steady state sorafenib C_max_ was determined as the maximum concentration following the final dose at 9:00 p.m. on Day 14.

Associations between physiological and molecular characteristics and sorafenib log transformed AUC and C_max_ were evaluated by univariate and multivariate linear regression. Continuous variables were checked for normality and non-linearity of association, sex was coded as a binary variable. A multivariable linear regression model to predict the log transformed sorafenib C_max_ was developed by stepwise forward inclusion of individually significant characteristics identified in the univariable regression analysis based on a probability of F to enter ≤ 0.05. The multivariable model (MIDS) predicted C_max_ was determined by back transformation of the model predicted log transformed C_max_. The capacity of MIDS to identify subjects with a sub-therapeutic simulated sorafenib C_max_ determined by scaling the reported threshold for simulation accuracy was evaluated using classification matrix analysis and is summarised as model sensitivity and specificity. The predictive performance of MIDS was assessed by receiver operating characteristic curve (ROC) analysis. Statistical analysis was conducted using R version 4.0.2 and IBM SPSS Statistics for Windows version 23 (Release 2015, IBM, Armonk, NY, USA).

### 4.7. Impact of Dose Individualisation

A simulation was conducted to evaluate the capacity of concentration-guided sorafenib dose adjustment to achieve a steady state sorafenib C_max_ within the range 4.78 to 5.78 μg/mL. Sorafenib exposure was simulated in a cohort of 500 subjects from the Sim-Cancer population (20 to 50 years old, 50% female) over 14 days with 400 mg sorafenib administered orally in a fasted-state every 12 h starting at 9:00 a.m. on Day 1. Sorafenib C_max_ was determined following the final dose at 9:00 p.m. on Day 14. Sorafenib exposure in subjects who failed to achieve a Day 14 normalised simulated C_max_ ≥ 4.78 µg/mL was simulated over an additional 14 days with sorafenib administered at a dose of 600 mg every 12 h starting at 9:00 a.m. on Day 15. The post dose adjustment sorafenib C_max_ was determined following the final dose at 9:00 p.m. on Day 28.

A simulation was conducted in the same cohort to evaluate the benefit of MIDS at baseline in conjunction with concentration-guided sorafenib dose adjustment. Demographic characteristics for the Sim-Cancer cohort were used to predict the normalised simulated Day 14 sorafenib C_max_ based on the multivariable model described previously. Based on MIDS subjects with a predicted sorafenib C_max_ ≥ 4.78 µg/mL received 400 mg sorafenib twice daily, while subjects with predicted sorafenib C_max_ < 4.78 µg/mL received 500 mg sorafenib twice daily. Sorafenib exposure was simulated over 28 days as described for concentration-guided sorafenib dose adjustment without MIDS, with C_max_ evaluated at Day 14 and Day 28 and a dose increase to 600 mg between Day 15 and Day 28 for individuals who failed to achieve a Day 14 C_max_ ≥ 4.78 µg/mL.

## Figures and Tables

**Figure 1 pharmaceuticals-14-00389-f001:**
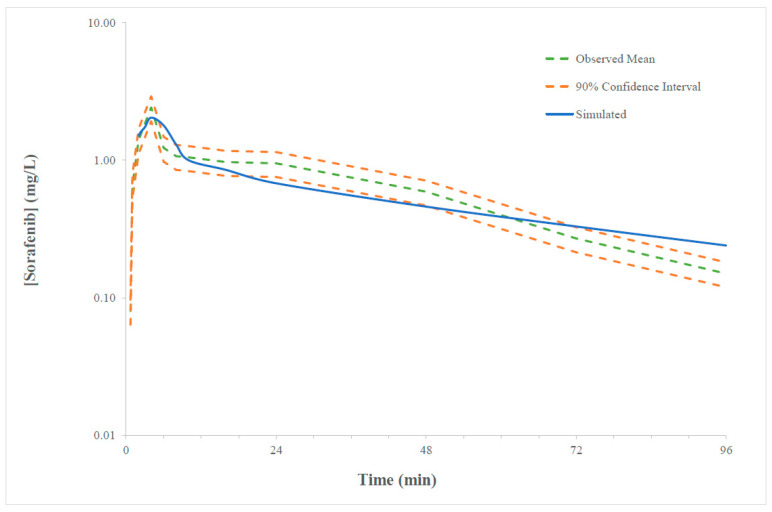
Representative overlay of simulated and observed (range) plasma concentration time curve of sorafenib (0–96 h) following 400 mg twice a day dosing. Solid blue line represented the mean model predicted exposure, dashed green line represented the mean observed exposure and dashed orange represented minimal and maximal 90% confidence intervals for the observed data.

**Figure 2 pharmaceuticals-14-00389-f002:**
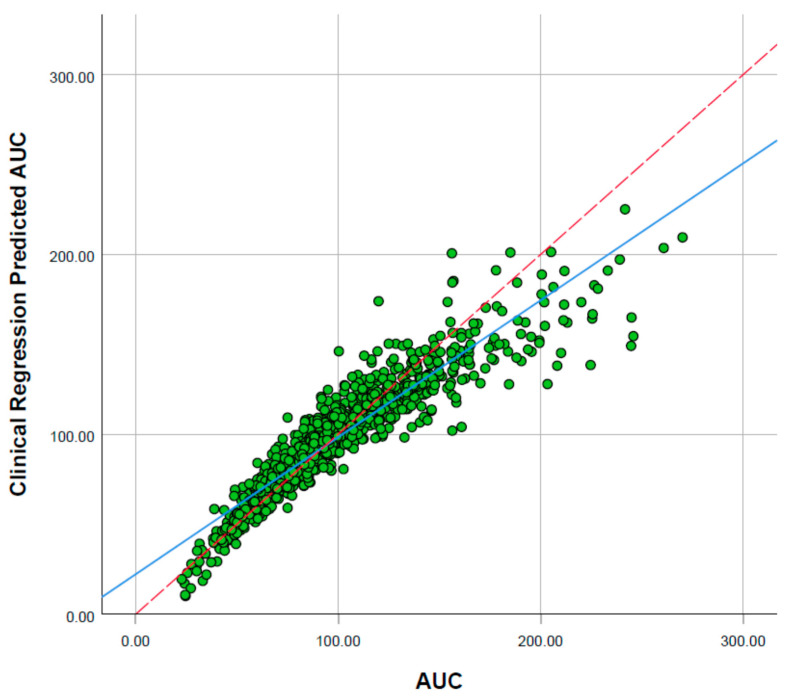
Correlation of model predicted steady-state sorafenib concentration predicted sorafenib AUC.

**Figure 3 pharmaceuticals-14-00389-f003:**
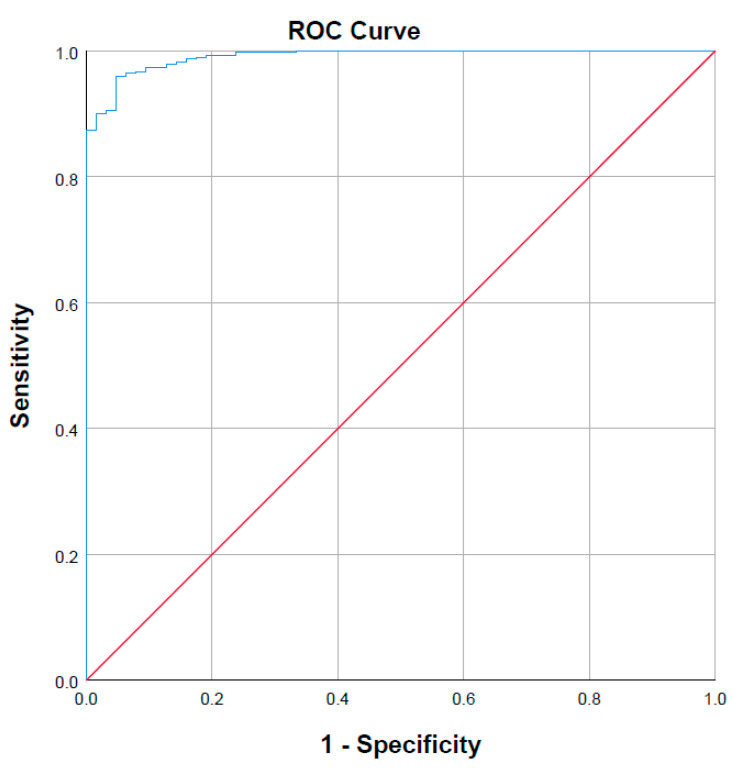
Receiver operating characteristic (ROC) curve demonstrating the prediction performance of a predicted steady state sorafenib AUC.

**Figure 4 pharmaceuticals-14-00389-f004:**
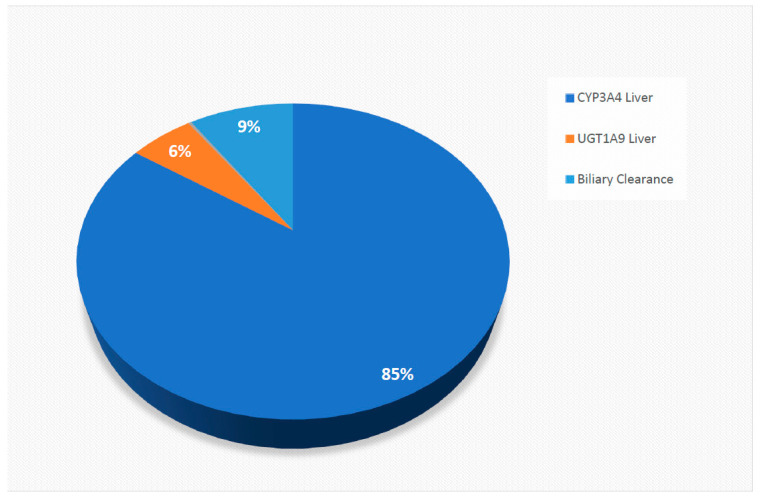
The pie chart demonstrated the relative contribution of CYP and UGT to simulated sorafenib elimination based on the predicted model.

**Table 1 pharmaceuticals-14-00389-t001:** Multivariable linear regression model performance characteristics.

Model	*R* ^2^	Std. Error of the Estimate	*R*^2^ Change	AUC ROC	AUC ROC Change
**a**	0.631	0.24141	0.631	0.953	0.953
**b**	0.781	0.18614	0.150	0.981	0.028
**c**	0.868	0.14458	0.087	0.990	0.009
**d**	0.873	0.14156	0.006	0.991	0.001
**e**	0.883	0.13619	0.010	0.991	-
**f**	0.883	0.13595	0.001	0.991	-

Model predictors (a) hepatic CYP3A4 abundance; (b) hepatic CYP3A4 abundance, albumin concentration; (c) hepatic CYP3A4 abundance, albumin concentration, BMI; (d) hepatic CYP3A4 abundance, albumin concentration, BMI, body surface area; (e) hepatic CYP3A4 abundance, albumin concentration, BMI, body surface area, sex; (f) hepatic CYP3A4 abundance, albumin concentration, BMI, body surface area, sex, weight; (g) hepatic CYP3A4 abundance, albumin concentration, BMI, body surface area, sex and weight.

**Table 2 pharmaceuticals-14-00389-t002:** Classification matrixes describing the capability of linear regression models to identify individuals with a sub-therapeutic sorafenib steady state C_max._

		Predicted Therapeutic C_max_		Percentage Correct
		Sub-Therapeutic	Therapeutic	
**Observed Therapeutic C_max_**	Sub-therapeutic	60 (true negative)	3 (false negative)	95.2
	Therapeutic	47 (false positive)	890 (true positive)	95.0

**Table 3 pharmaceuticals-14-00389-t003:** Number of participant below, within and above target concentration range with different sorafenib dosing protocols.

Dosing Protocol	Day 14			Day 28		
	<4.78 µg/mL	4.78 to 5.78 µg/mL	>5.78 µg/mL	<4.78 µg/mL	4.78 to 5.78 µg/mL	>5.78 µg/mL
Flat dosing	62	116	322	62	116	322
Concentration-guided dosing	62	116	322	5	130	365
Concentration-guided dosing with MIDS	34	135	331	5	164	336

**Table 4 pharmaceuticals-14-00389-t004:** Model inputs used to build the sorafenib compound model.

Parameter	Value	Source
**Physicochemical properties** Molecular weight Log P_o:w_ Hydrogen bond donor Species	464.82 g/mol 4.54 3 Base	[43] [43]
**Protein binding** B/P f_up_	0.55 0.0048	[43] [43]
**Absorption (ADAM model)** f_a_ k_a_ (L/h)	0.99 1.75	Predicted Predicted
**Permeability** P_eff_, man (10^−4^ cm/s) Caco-2 (10^−6^ cm/s)	4.01 24.1	Predicted
**Formulation** Solid formulation	Immediate release	[43]
**In vivo pharmacokinetic properties (full PBPK model)** Prediction model K_p_ scalar	1 0.7	Predicted
**CYP metabolism: ISEF adjusted recombinant enzyme kinetics (CL_int_; μL/min/pmol)** CYP3A4	2.6	[18]
**UGT metabolism: ISEF adjusted recombinant enzyme kinetics (CL_int_; μL/min/mg)** UGT1A9	20.1	[18]

## Data Availability

The data that support the findings of this study are available from the corresponding author upon reasonable request.

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
