# Peer review of "Mechanistic Modelling Identifies and Addresses the Risks of Empiric Concentration-Guided Sorafenib Dosing"

_pharmaceuticals, 2021, doi:10.3390/ph14050389_

Round 1

Reviewer 1 Report

The manuscript is interesting, however some issues should be clarified:

  • Did this research require the agreement of the local ethic committee
  • Were there any inclusion or exclusion criteria to the study?

Author Response

Point 1: Did this research require the agreement of the local ethic committee

Response 1: We are pleased to clarify to the Reviewer that as this study was conducted using only in silico methods there was no requirement for approval by any local human or animal ethics committee.

Point 2: Were there any inclusion or exclusion criteria to the study?

Response 2: The virtual population used in the studies performed to evaluate the impact of dose optimisation included an equal distribution of male and female cancer patients aged 20 to 50 years – these were the effective inclusion criteria for the virtual trials. No further inclusion or exclusion criteria were applied.

Reviewer 2 Report

The manuscript titled “mechanistic modelling identifies and addresses the risks of empiric concentration guided sorafenib dosing” attempted to assess the sorafenib dosing using an in silico software.

As such the manuscript is well written. However, there are some deficiencies in the manuscript which need to be addressed to make it publishable:

  1. Were physiological/pathological conditions of the cancer patients considered in building the PBPK models?
  2. Virtual stratification of patients will be useful.
  3. Authors need to include Parameter Sensitivity Analysis (PSA) data to validate the model.
  4. Typically, cancer patients are on multiple medications. How the calculated sorafenib doses are affected by medications co-administered to cancer patients?
  5. Page 1, line 8: there is a typo. “….this study was to evaluate……"

Author Response

Point 1: Were physiological/pathological conditions of the cancer patients considered in building the PBPK models?

Response 1: We thank the Reviewer for this insightful question. The physiological and pathological characteristics of the Sim-cancer population have been determined based on a meta-analysis of cancer patients enrolled in clinical trials. A brief description of these characteristics and a reference (new ref 47) to the original manuscript describing the population has been added to the methods section of the manuscript (line 286 - 288).

Point 2: Virtual stratification of patients will be useful.

Response 2: The Reviewer makes an excellent conceptual point. Fortunately, the PBPK platform utilised in the current analysis allows for the consideration of effective parallel universes whereby an identical population can undertake each arm of the study at the same time in parallel. The benefit of this study design is that all variability in outcome that is not associated with the intervention is effectively removed. As such there is no need for virtual stratification of the study arms.

Point 3: Authors need to include Parameter Sensitivity Analysis (PSA) data to validate the model.

Response 3: As requested by the Reviewer, the additional statements have been made (line 95 - 98). The parameter sensitivity analysis figure has been provided in Supplementary Figure 1.

Point 4: Typically, cancer patients are on multiple medications. How the calculated sorafenib doses are affected by medications co-administered to cancer patients?

Response 4: The Reviewer makes an excellent point. Unfortunately, while Simcyp is ideally equipped to evaluate the impact of specific Drug-Drug Interactions (DDIs), the platform is not well suited to the consideration of the impact of diverse and sporadic concomitant medication regiments. Alternate approaches such as secondary analysis of clinical trial data (see Ruanglertboon, W., et al. (2020). "The effect of proton pump inhibitors on survival outcomes in advanced hepatocellular carcinoma treated with sorafenib." J Cancer Res Clin Oncol 146(10): 2693-2697) are far better suited to evaluate such effects. As such, while valuable this question was beyond the scope of the current study.

Point 5: Page 1, line 8: there is a typo. “….this study was to evaluate……"

Response 5: We thank the Review for identifying this typo and have make the suggested correction (line 8).

Reviewer 3 Report

1. "has been demonstrated to have no impact on survival outcomes in HCC patients 39 treated with sorafenib." No information on RCC available.
2.
Why did the authors omit the active metabolite of sorafenib in their analysis? 3.Why did the authors take into account the level of albumin, since the drug is bound to albumin in 99.8%. And the authors' analysis lacks the total fraction of the drug. It would be worth including the analysis, where the authors determined the free fraction of the drug and made a correlation with the level of albumin.  

Author Response

Point 1: "has been demonstrated to have no impact on survival outcomes in HCC patients 39 treated with sorafenib." No information on RCC available.

Response 1: As requested by the Reviewer, this information has been added to the manuscript, including two new references [new reference 9 and 10] (line 39 and 40).

- Reference 9 is “Proton Pump Inhibitors and Survival Outcomes in Patients With Metastatic Renal Cell Carcinoma”.

- Reference 10 is “Effect of concomitant proton pump inhibitor (PPI) on effectiveness of tyrosine kinase inhibitor (TKI) in patients with metastatic renal cell carcinoma (mRCC)”.

These two references provided the information supporting that using proton pump inhibitors concomitantly with sorafenib had no impact on survival outcomes in RCC patients.

Point 2: Why did the authors omit the active metabolite of sorafenib in their analysis?

Response 2: The concentration thresholds used in the current analysis, which have also been broadly quoted with respect to monitoring sorafenib by therapeutic drug monitoring (i.e. Cmax > 4.78 μg/mL) is based purely on consideration of the parent compound. As such, in order to directly compare against published in vivo data, the current study similarly focused solely on the parent compound.

Point 3: Why did the authors take into account the level of albumin, since the drug is bound to albumin in 99.8%. And the authors' analysis lacks the total fraction of the drug. It would be worth including the analysis, where the authors determined the free fraction of the drug and made a correlation with the level of albumin.

Response 3: We agree with the Reviewer that albumin concentration is a potential key determinant of sorafenib exposure. As requested we have included an additional analysis correlating albumin concentration with total and unbound sorafenib concentration and unbound sorafenib. As can be observed from Supplementary Figure 2, sorafenib unbound fraction decreases exponentially with increased albumin concentration.

Reviewer 4 Report

This manuscript describes the risk of empiric concentration of sorafenib dosing based on mechanistic modeling. My comments to this manuscript are as followings:

  1. All models used in this manuscript should be presented with the compared statistical values together.
  2. Please describe the detail process to set up the mechanistic models and process to predict the results for the reproducible modeling.

Author Response

Point 1: All models used in this manuscript should be presented with the compared statistical values together.

Response 1: We thank the Reviewer for commenting on the manuscript and appreciate the need for improved clarity on comparative statistics. We have proof read the manuscript and updated where appropriate. Chiefly, Table 1 has been updated to indicate the statistical differences in ROC AUCs between multivariable model increments (line 135).

Table 1. Multivariable linear regression model performance characteristics.

Model

R2

Std. Error of

the Estimate

R2 Change

AUC ROC

AUC ROC change

a

0.631

0.24141

0.631

0.953

0.953

b

0.781

0.18614

0.150

0.981

0.028

c

0.868

0.14458

0.087

0.990

0.009

d

0.873

0.14156

0.006

0.991

0.001

e

0.883

0.13619

0.010

0.991

-

f

0.883

0.13595

0.001

0.991

-

Point 2: Please describe the detail process to set up the mechanistic models and process to predict the results for the reproducible modeling.

Response 2: We respectfully draw the Reviewer’s attention to section ‘4. Materials and Methods’ of the methods where the process for undertaking the simulations and modelling was outlined.